

# Neurodevelopmental problems and extremes in BMI

Nóra Kerekes[1,2], Armin Tajnia[1], Paul Lichtenstein[3], Sebastian Lundström[1,2,4], Henrik Anckarsäter[1], Thomas Nilsson[1] and Maria Råstam[5]

[1] CELAM (Centre for Ethics, Law and Mental Health), Institute of Neuroscience and Physiology, University of Gothenburg, Gothenburg, Sweden
[2] Swedish Prison and Probation Services, R&E, Sweden
[3] Department of Medical Epidemiology and Biostatistics, Karolinska Institutet, Stockholm, Sweden
[4] Gillberg Neuropsychiatry Centre, Institution of Neuroscience and Physiology, University of Gothenburg, Gothenburg, Sweden
[5] Department of Clinical Sciences, Lund University, Lund, Sweden

Corresponding author
Nóra Kerekes,
nora.kerekes@neuro.gu.se

## ABSTRACT

**Background.** Over the last few decades, an increasing number of studies have suggested a connection between neurodevelopmental problems (NDPs) and body mass index (BMI). Attention deficit/hyperactivity disorder (ADHD) and autism spectrum disorders (ASD) both seem to carry an increased risk for developing extreme BMI. However, the results are inconsistent, and there have been only a few studies of the general population of children.

**Aims.** We had three aims with the present study: (1) to define the prevalence of extreme (low or high) BMI in the group of children with ADHD and/or ASDs compared to the group of children without these NDPs; (2) to analyze whether extreme BMI is associated with the subdomains within the diagnostic categories of ADHD or ASD; and (3) to investigate the contribution of genetic and environmental factors to BMI in boys and girls at ages 9 and 12.

**Method.** Parents of 9- or 12-year-old twins ($n = 12,496$) were interviewed using the Autism—Tics, ADHD and other Comorbidities (A-TAC) inventory as part of the Child and Adolescent Twin Study in Sweden (CATSS). Univariate and multivariate generalized estimated equation models were used to analyze associations between extremes in BMI and NDPs.

**Results.** ADHD screen-positive cases followed BMI distributions similar to those of children without ADHD or ASD. Significant association was found between ADHD and BMI only among 12-year-old girls, where the inattention subdomain of ADHD was significantly associated with the high extreme BMI. ASD scores were associated with both the low and the high extremes of BMI. Compared to children without ADHD or ASD, the prevalence of ASD screen-positive cases was three times greater in the high extreme BMI group and double as much in the low extreme BMI group. Stereotyped and repetitive behaviors were significantly associated with high extreme BMIs.

**Conclusion.** Children with ASD, with or without coexisting ADHD, are more prone to have low or high extreme BMIs than children without ADHD or ASD.

## INTRODUCTION

A recent report estimated that 43 million children worldwide are either overweight or obese, and another 92 million are at risk of becoming overweight (*de Onis, Blossner & Borghi, 2010*). The same study also showed that the percentage of overweight children has increased from 4% in 1990 to almost 7% in 20 years (1990–2010). In Sweden, from 1984 to 2000, the number of children with a body mass index (BMI) indicating obesity increased fourfold and the number of children who were overweight doubled (*Marild et al., 2004*). In a recent Swedish study of children aged 7 to 9 years, 17% were overweight and another 3% were obese (*Sjoberg et al., 2011*).

Equally important is the recognition of underweight in children; moderate to severe low BMI carries the risk of medical instability and long-term medical consequences (*Hudson et al., 2012*; *Arcelus et al., 2011*). The incidence of restrictive eating and low BMI in children has been estimated at 3.1 per 100,000 person-years in the United Kingdom (*Nicholls, Lynn & Viner, 2011*) and 2.6 per 100,000 person-years in Canada (*Pinhas et al., 2011*).

Many studies of twins and families have quantified the contributions of genetic and environmental factors to inter-individual differences in BMI. They were summarized in two reviews (*Elks et al., 2012*; *Maes, Neale & Eaves, 1997*), which reported great heterogeneity of the results (BMI heritability ranging between 0.47 and 0.90)—a variation that could be partly explained by variations in factors such as age in the different study populations (*Elks et al., 2012*).

Besides BMI, in the present study we aim to focus on mental health, as measured by the diagnoses of attention deficit/hyperactivity disorder (ADHD) and the broadly defined autism spectrum disorders (ASD) (*Boyle et al., 2011*; *Charman, 2011*). ADHD is recognized by signs of inattention, hyperactivity, and impulsivity (American Psychiatric Association *APA, 2013*), while ASD is characterized by deficits in social communication and behavioral flexibility (*APA, 2013*). Over the last few decades, the number of schoolchildren diagnosed with ADHD has increased from 2%–5% (*Kadesjo & Gillberg, 1998*) to 7%–9% (*Scahill & Schwab-Stone, 2000*). In recent decades, more children have also been diagnosed with the broader spectrum of ASD (*Fombonne, 2009*). The overlap between these two forms of neurodevelopmental disorder (NDP) is substantial, reaching 65%–80% (*Holtmann, Bolte & Poustka, 2005*; *Lee & Ousley, 2006*). Twin studies suggest genetic and enviromental factors common to both disorders (*Lichtenstein et al., 2010*).

Of the few studies that have attempted to analyze the association between BMI and NDPs, most have been conducted in mainly adult populations. While it is not possible to draw a general conclusion from these studies, they have found a tendency for an overlap between overweight and ADHD or ASD (*Agranat-Meged et al., 2005*; *Altfas, 2002*; *de Zwaan et al., 2011*; *Dubnov-Raz, Perry & Berger, 2011*; *Kim et al., 2011*). A recent study (including over 12,000 Swedish 9- or 12-year-old twins) found a 0.6% prevalence of restrictive eating as defined by parent-reported weight stop or loss combined with the child's fear of gaining weight (*Rastam et al., 2013*). Interestingly, in the group with eating problems there was a clear overrepresentation of children with symptoms of ADHD and/or ASD (*Rastam et al., 2013*).

In the present report, we aimed to pursue this finding by analyzing the association between extreme (low or high) BMI and the presence of ADHD and or ASD using the data collected in one of today's largest child twin studies. Because we used cross-sectional data, we did not attempt to disentangle causality between these factors.

The specific aims of the present study were to investigate:

(1) the prevalence of extreme BMI in the group of children with ADHD and/or ASD;
(2) whether extreme BMI is associated with the subdomains within the diagnostic categories of ADHD or ASD; and
(3) the contribution of genetic and environmental factors to BMI in boys and girls at ages 9 and 12.

## SUBJECTS AND METHODS

### Study population

The study population was recruited from the ongoing Child and Adolescent Twin Study in Sweden (CATSS), a longitudinal, nationwide study focusing on mental health problems during childhood and adolescence (*Anckarsater et al., 2011*). CATSS is based on telephone interviews with the parents of all 9-year-old twins in Sweden. (During the first three years of the study, parents of all 12-year-olds were also included (CATSS-9/12)). In the present study, we analyzed the data on twins born between 1992 and 1998 ($n = 12{,}496$) who participated in the CATSS-9/12. We excluded 407 children with congenital or early brain damage, known chromosomal syndromes, diabetes, cancer, celiac disease, or other defined medical conditions with potential weight effect, and another 734 (6% of those who participated in the study) because of missing data. The final study population thus consisted of 11,355 children, with close to equal distribution between boys and girls (51.9% and 48.1%, respectively) and 9- and 12-year-olds (47.6% and 52.4%, respectively). Distribution of zygosities in the study was 27% monozygotic (MZ) twins, 35% dizygotic (DZ)–same sex (ss), and 33% DZ–different sex; 5% of the children could not be typed by zygosity. Only twin pairs with complete data on weight, height, and symptoms of NPDs were included in calculating heritability estimates; 1518 MZ and 1900 DZss twin pairs were eligible for inclusion.

### Measures

#### BMI

BMI is calculated as weight in kg/height in m$^2$. For children, the use of BMI as a tool for assessing body fat is more complex than for adults. BMI references for children are age- and gender-specific, and the cutoffs for underweight, overweight, and obesity are population-specific. Many countries have their own population references in addition to the well-known international references of the World Health Organization (*WHO, 2007*). In both clinical practice and research the *Karlberg, Luo & Albertsson-Wikland (2001)* BMI reference is used as the standard in Sweden. The Karlberg BMI reference was derived from an original study of 3650 healthy singletons born between 1973 and 1975 in the western part of Sweden. Their mean BMI $\pm$ 1, 2, and 3 standard deviations (SD) were calculated in addition to

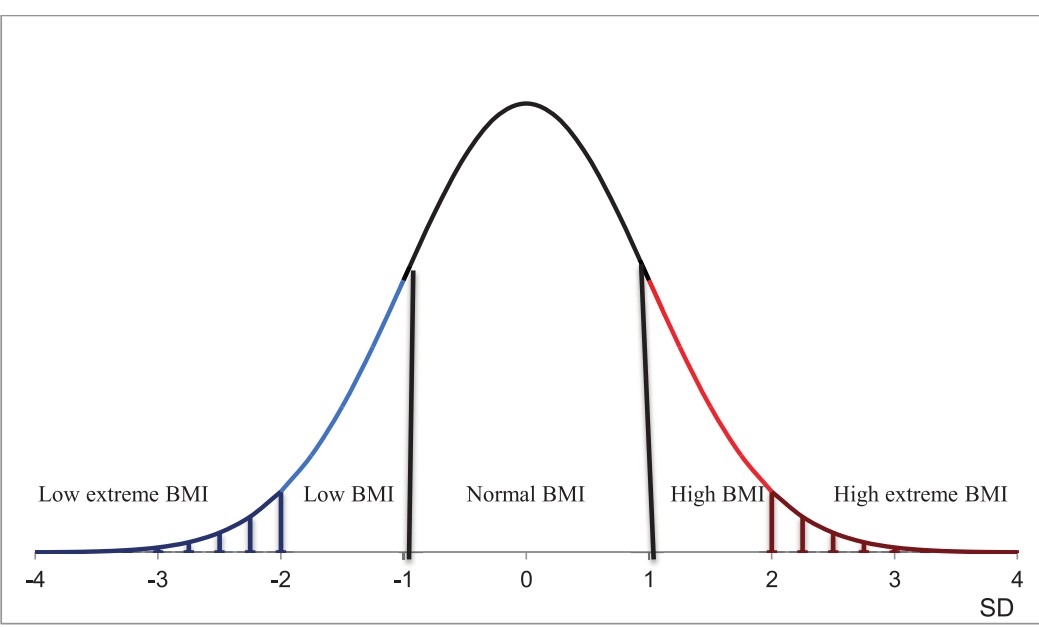

SD = Standard deviation

**Figure 1 Graphical presentation of BMI categories.** Low extreme BMI was defined as more than 2SD below the mean, low BMI as more than 1SD but less than 2SD below the mean, normal BMI as within the range of minus one and plus 1SD from the mean, high BMI as more than 1SD but not more than 2SD above the mean, and high extreme BMI as more than 2SD above the mean.

previous percentile charts (*WHO, 2007*) to create a better tool for clinical evaluation (*Reilly & Kelly, 2011*). Given the geographic origin and the age range of the present study cohort, only data representing the 9-year-old ($n = 828$ boys; $n = 790$ girls) and 12-year-old ($n = 640$ boys; 676 girls) children from the Karlberg study (*Karlberg, Luo & Albertsson-Wikland, 2001*) were used as BMI references for comparison with the CATSS twins.

In the present study, information about children's weight and height was reported by their parents. To avoid bias, we constructed a new reference for BMI because (1) we had a much larger database than those used in the WHO and the Karlberg studies, (2) our sample represented twins, not singletons, and (3) our data include newer generations of children. BMI scores were transformed with the help of Box–Cox transformations to ensure normal distribution (for a detailed statistical description, see Appendix S1) and then converted to BMI standard deviation scores. In the present study, low extreme BMI was defined as more than 2SD below the mean, low BMI as more than 1SD but less than 2SD below the mean, normal BMI as within the range of minus one and plus 1SD from the mean, high BMI as more than 1SD but not more than 2SD above the mean, and, finally, high extreme BMI as more than 2SD above the mean (Fig. 1).

### The Autism–Tics, ADHD and other comorbidities inventory

The Autism–Tics, ADHD and other Comorbidities (A-TAC) inventory was developed for CATSS as a screening instrument to identify most major child and adolescent psychiatric conditions by evaluating parental information about children, collected during a telephone interview. The 96 A-TAC items reflect diagnostic criteria listed in the *Diagnostic and*
**Table 1 Age and gender distribution in the study population of screen positive and comparison groups.**

|  | Number of boys (%) | | Number of girls (%) | |
| --- | --- | --- | --- | --- |
|  | Age 9 | Age 12 | Age 9 | Age 12 |
| Comparison group | 2449 (86.9) | 2630 (85.5) | 2412 (93.3) | 2689 (93.6) |
| ADHD group | 263 (9.3) | 303 (9.8) | 144 (5.6) | 122 (4.2) |
| ASD group | 24 (0.9) | 31 (1.0) | 4 (0.2) | 22 (0.8) |
| ADHD + ASD group | 83 (2.9) | 113 (3.7) | 25 (1.0) | 41 (1.4) |
| Study population | 2819 (100) | 3077 (100) | 2585 (100) | 2874 (100) |

Notes.
ADHD, attention deficit/hyperactivity disorder; ASD, autism spectrum disorder.

*Statistical Manual of Mental Disorders*, 4th ed. (*APA, 1994*) and well-known clinical features of various disorders. Defined modules can be clustered into domains corresponding to the main problem areas of specific diagnoses, for example the *concentration/attention* and *impulsiveness/activity* modules form the ADHD domain, while the modules *language*, *social interaction*, and *flexibility* form the ASD domain. The psychometric properties of the ADHD and ASD domains have been described and validated (*Hansson et al., 2005*; *Larson et al., 2010*) and found to have excellent predictive properties, with areas under the curve (AUC) of 0.94 for ADHD and 0.96 for ASD. For screening purposes, the cutoffs of $\geq 6.0$ (of the maximum 19 points) for ADHD and $\geq 4.5$ (of the maximum 17 points) for ASD were previously defined, validated, and described to have excellent or good sensitivity and specificity (0.91 and 0.73 respectively for ADHD and 0.91 and 0.80 for ASD; *Larson et al., 2010*).

In the present study, children scoring $\geq 6.0$ in the ADHD domain will be referred to as the ADHD group, children scoring $\geq 4.5$ in the ASD domain as the ASD group, and those fulfilling criteria for both ADHD and ASD as the ADHD + ASD group. The children not fulfilling criteria for ADHD or ASD constitute the comparison group (Table 1).

Gender and age were close to equally distributed in the four groups. However, since NDPs were twice as frequent in boys as in girls, there were slightly more girls in the comparison group.

## Medication

Overall, parents reported 17 different prescription drugs used by children either daily or during the month prior to the interview; only three of these (betamethasone, methylphenidate, and desmopressin) had known effects on weight. In the total study population less than 1% ($n = 95$) received medication with potential influence on weight, and over half of these ($n = 48$) did not screen positive for ADHD or ASD. Only one of the 81 children who screened positive for ASD, 20 of the 832 who screened positive for ADHD, and 26 of the 262 children who screened positive for ADHD and ASD combined had medication. Of the 95 children who were on potentially weight-modulating medication, 93% had a normal BMI, 3% had an extremely high BMI, and 4% had an extremely low BMI.

*Data analysis*

A chi-square test of independence was used to analyze any dependence between the ADHD, ASD, ADHD + ASD, and comparison groups and the low extreme, low, high, and high extreme BMI categories. A post hoc test of the standardized residuals was applied to evaluate which of these groups were major contributors to a significant chi-square test result.

Binary logistic regression models fitted to generalized estimated equation (GEE) models were used to estimate the association between the dependent variable, the BMI categories (low extreme to normal, low to normal, normal to high, and normal to high extreme) and the independent variables, ADHD and ASD scores as covariates, and age and gender as cofactors. First, univariable models were applied to study the separate effects of ADHD and ASD on BMI; then multivariable models were used to investigate the effects of coexisting ADHD and ASD on BMI. These GEE models take the statistical dependence within twin pairs into consideration. When a significant effect of ADHD or ASD scores was measured, new GEE models were fitted to investigate the detailed contribution of subdomains within the domain. In these models, the scores on activity/impulsiveness and attention (ADHD dyad) and social interaction, flexibility, and language (ASD triad) could be used as independent covariates.

Twin modeling can help to disentangle genetic from environmental effects based on the difference in genetic relatedness between MZ and DZ twins. MZ twins are genetically identical while DZ twins share, on average, half of their segregated alleles. Comparisons between MZ and DZ twin pairs can indicate whether the examined traits are affected by genetic factors. In this study intra-class correlations were calculated separately for MZ and DZ twins and univariate structural equation modeling was performed using Mx model-fitting software. In the ACE model, genetic factors contributing to a trait are typically denoted as "A" (additive genetics) components, while environmental factors can be classified either as "C" (common environmental factors making the twins similar) or "E" (unique environmental factors making the twins dissimilar).

*Ethical considerations*

The CATSS study complies with the Helsinki Declaration and has been granted ethical approval by the Karolinska Institutet Ethical Review Board (Dnr 03-672 and 2010/507-31/1). All subjects participated under an informed consent procedure. All data analyses were performed on anonymized data.

# RESULTS

## Prevalence of ADHD and ASD diagnoses according to BMI categories

There was a normal distribution of BMI in the comparison group, with 69% of the children having a normal BMI and close to equal distribution of the lower and higher ends of the scale (Fig. 2). The BMI distribution in the ADHD group was similar to the comparison group. However, the ASD group showed almost three times greater prevalence of the high extreme BMI (standardized residual = 2.3) and nearly twice as great a prevalence of low

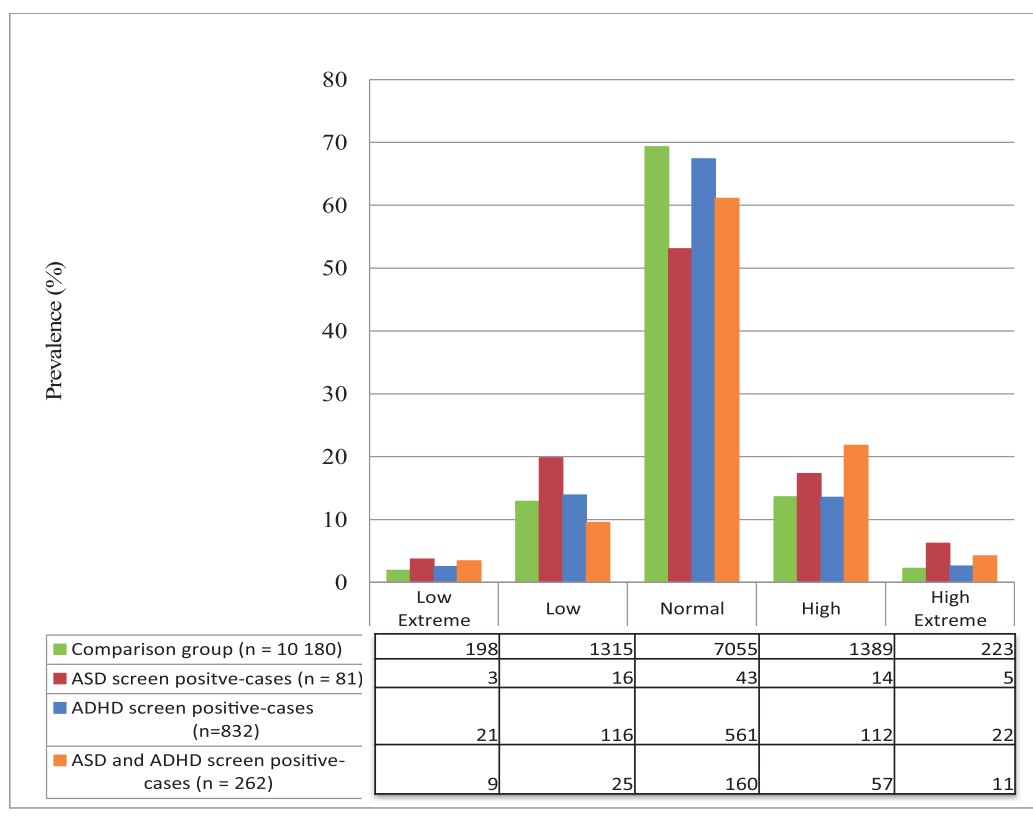

| | Low Extreme | Low | Normal | High | High Extreme |
|---|---|---|---|---|---|
| Comparison group (n = 10 180) | 198 | 1315 | 7055 | 1389 | 223 |
| ASD screen positve-cases (n = 81) | 3 | 16 | 43 | 14 | 5 |
| ADHD screen positive-cases (n=832) | 21 | 116 | 561 | 112 | 22 |
| ASD and ADHD screen positive-cases (n = 262) | 9 | 25 | 160 | 57 | 11 |

ADHD = attention deficit/ hyperactivity disorder; ASD = autism spectrum disorder

**Figure 2** **Prevalence of NDPs and their distribution between low extreme, low, normal, high, and high extreme categories of BMI.**

extreme BMI (standardized residual = 1.1), with only about half having a normal BMI (Fig. 2). The ADHD + ASD group showed a roughly two-fold higher prevalence in the high extreme category (standardized residual = 2.0), a significantly increased prevalence in the high category (standardized residual = 3.4), and an increased, but not significantly, prevalence in the low extreme BMI category (Fig. 2) over the comparison group. In general, the NDP groups had higher prevalences than the comparison group in all BMI categories other than normal ($X^2$ (12, $N = 11,355$) = 39.941, $P < 0.00$).

## Association between extremes in BMI and NDPs

Both ADHD and ASD modules were weakly but significantly associated with high and high extreme BMI in the univariable models. In the multivariable model (including both ADHD and ASD as covariates) only the ASD module kept its significant positive association with these dependent variables (Table 2).

When measuring the association of the subdomains in univariable models, problems with attention from the ADHD domain and problems with social interaction and flexibility from the ASD domain increased the risk for high BMI. Only the social interaction subdomain kept its significant positive association with high BMI in a

**Table 2 Association between BMI categories and NDPs (GEE models).**

| Variable | Crude measure | | | | | Univariable model | | Multivariable model[a] | |
|---|---|---|---|---|---|---|---|---|---|
| | *n* | Min | Max | Mean | SD | OR | 95% CI | OR | 95% CI |
| **High extreme BMI** | | | | | | | | | |
| ADHD | 8,080 | 0.0 | 19.0 | 1.87 | 2.92 | **1.06**[***] | 1.03–1.09 | 1.01 | 0.97–1.05 |
| ASD | 8,080 | 0.0 | 17.0 | 0.72 | 1.47 | **1.14**[***] | 1.09–1.20 | **1.34**[**] | 1.06–1.22 |
| Inattention | 8,079 | 0.0 | 9.0 | 0.98 | 1.67 | **1.13**[***] | 1.07–1.19 | **1.16**[***] | 1.08–1.25 |
| Activity/impulsiveness | 8,080 | 0.0 | 10.0 | 0.88 | 1.58 | 1.05 | 1.00–1.11 | 1.00 | 0.88–1.04 |
| Social interaction | 8,071 | 0.0 | 6.0 | 0.24 | 0.58 | **1.37**[***] | 1.21–1.56 | 1.13 | 0.98–1.44 |
| Flexibility | 8,080 | 0.0 | 5.0 | 0.23 | 0.57 | **1.47**[***] | 1.27–1.71 | **1.36**[*] | 0.06–1.75 |
| Language | 8,076 | 0.0 | 6.0 | 2.24 | 0.58 | **1.28**[**] | 1.09–1.51 | 0.98 | 0.74–1.29 |
| **High BMI** | | | | | | | | | |
| ADHD | 9,391 | 0.0 | 19.0 | 1.88 | 2.95 | **1.02**[*] | 1.00–1.04 | 1.01 | 0.98–1.03 |
| ASD | 9,391 | 0.0 | 17.0 | 0.73 | 1.48 | **1.06**[**] | 1.02–1.09 | **1.05**[*] | 1.01–1.10 |
| Inattention | 9,390 | 0.0 | 9.0 | 0.98 | 1.68 | **1.03**[*] | 1.00–1.07 | 1.02 | 0.99–1.06 |
| Activity/impulsiveness | 9,391 | 0.0 | 10.0 | 0.89 | 1.60 | 1.03 | 0.99–1.07 | 1.02 | 0.98–1.06 |
| Social interaction | 9,378 | 0.0 | 6.0 | 0.25 | 0.59 | **1.16**[***] | 1.07–1.26 | **1.14**[*] | 1.02–1.28 |
| Flexibility | 9,391 | 0.0 | 5.0 | 0.23 | 0.57 | **1.15**[**] | 1.05–1.26 | 1.10 | 0.98–1.25 |
| Language | 9,387 | 0.0 | 6.0 | 0.24 | 0.58 | 1.06 | 0.97–1.17 | 0.94 | 0.83–1.06 |
| **Low BMI** | | | | | | | | | |
| ADHD[b] | 9,291 | 0.0 | 19.0 | 1.84 | 2.92 | 1.00 | 0.98–1.02 | 1 | 0.97–1.02 |
| ASD[b] | 9,291 | 0.0 | 17.0 | 0.70 | 1.43 | 1.02 | 0.98–1.05 | 1.02 | 0.98–1.07 |
| **Low extreme BMI** | | | | | | | | | |
| ADHD[b] | 8,050 | 0.0 | 19.0 | 1.85 | 2.93 | 1.03 | 0.99–1.06 | 1 | 0.95–1.05 |
| ASD | 8,050 | 0.0 | 17.0 | 0.71 | 1.46 | **1.07**[*] | 1.01–1.34 | 1.06 | 0.97–1.16 |
| Social interaction | 8,041 | 0.0 | 6.0 | 0.24 | 0.58 | 1.08 | 0.90–1.29 | 0.86 | 0.68–1.08 |
| Flexibility | 8,050 | 0.0 | 5.0 | 0.22 | 0.56 | **1.24**[*] | 1.01–1.36 | **1.27**[*] | 1.02–1.58 |
| Language | 8,046 | 0.0 | 6.0 | 0.24 | 0.57 | **1.17**[*] | 1.01–1.36 | 1.10 | 0.92–1.33 |

**Notes.**

[*] $p < 0.05$.

[**] $p < 0.01$.

[***] $p < 0.001$.

[a] Adjusted for sex and age.

[b] No significant effects of the variable were measured and therefore no new GEE models were fitted for the subdomains within the domain.

ADHD, attention deficit/hyperactivity disorder; ASD, autism spectrum disorder.

multivariable model. All three ASD subdomains were significantly associated with high extreme BMI in univariable models, while in the multivariable model only the subdomain measuring problems with flexibility retained a significant relation (Table 2). Among the ADHD subdomains, only inattention increased the risk of high extreme BMI in both univariable and multivariable models.

Neither ADHD nor ASD were associated with low BMI, but ASD, in a univariable model, had a weak positive association with low extreme BMI. The flexibility and language subdomains were associated with this risk, but only the flexibility subdomain retained its significant effect in a multivariable model.

**Table 3  Intraclass correlations and ACE models of BMI in age and gender selected groups.**

| Intraclass correlation (95% CI) | | | | | | | |
|---|---|---|---|---|---|---|---|
| 9-year-old boys | | 12-year-old boys | | 9-year-old girls | | 12-year-old girls | |
| MZ *n* = 357 | DZ *n* = 498 | MZ *n* = 409 | DZ *n* = 537 | MZ *n* = 361 | DZ *n* = 392 | MZ *n* = 391 | DZ *n* = 473 |
| 0.85 | 0.57 | 0.88 | 0.45 | 0.86 | 0.57 | 0.91 | 0.55 |
| (0.82–0.88) | (0.50–0.62) | (0.85–0.90) | (0.38–0.51) | (0.83–0.89) | (0.50–0.64) | (0.89–0.94) | (0.48–0.61) |

| Estimates of genetic and environmental effects (95% CI) | | | | | | | | | | | |
|---|---|---|---|---|---|---|---|---|---|---|---|
| 9-year-old boys | | | 12-year-old boys | | | 9-year-old girls | | | 12-year-old girls | | |
| A | C | E | A | C | E | A | C | E | A | C | E |
| 0.57 | 0.28 | 0.15 | 0.87 | 0.01 | 0.12 | 0.55 | 0.31 | 0.14 | 0.77 | 0.15 | 0.08 |
| (0.46–0.69) | (0.16–0.39) | (0.13–0.17) | (0.75–0.90) | (0.00–0.13) | (0.10–0.14) | (0.43–0.68) | (0.17–0.42) | (0.12–0.17) | (0.65–0.89) | (0.02–0.26) | (0.07–0.10) |

**Notes.**

MZ, monozygotic twins; DZ, dizygotic twins; A, additive genetic effect; C, common environment; E, unique environment.

### Etiology of BMI

Intra-class correlations were measured in gender- and age-specific groups. They were stronger in MZ than in DZ twins (Table 3) in all groups. In the ACE models, the genetic effect was higher in 12-year-olds than in 9-year-olds in both girls and boys, and the effect of common environment was lower.

### Twins' BMI standard versus singletons'

Using the newly constructed Swedish BMI reference (specific to 9- and 12-year-old twins) for this study (Appendix S1), there were no significant differences in the mean BMI of boys or girls compared with Swedish (*Karlberg, Luo & Albertsson-Wikland, 2001*) or international (*WHO, 2007*) BMI references (Fig. 3). However, in our BMI reference, the cutoffs at +1SD, +2SD, and +3SD were higher in all study groups (with the exception of 12–year-old girls (Fig. 3D)), while at −1SD, −2SD, and −3SD, BMI was consequently lower than these BMI values in the Swedish or international BMI references, providing stricter criteria for extremes.

## DISCUSSION

*Major findings of the present study were:*

1. In the group of children with ASD (with or without ADHD), there was an increased prevalence of both high extreme and low extreme BMI.
2. High extreme BMI was associated with high scores on the inattention subdomain of ADHD, while both high and low extreme BMIs were associated with high scores on the subdomain of ASD measuring inflexibility.
3. The genetic effect on BMI was higher and the effect of shared environment was lower in 12-year-olds compared to 9-year-olds.
4. BMI of twins seems to be representative of the general population.

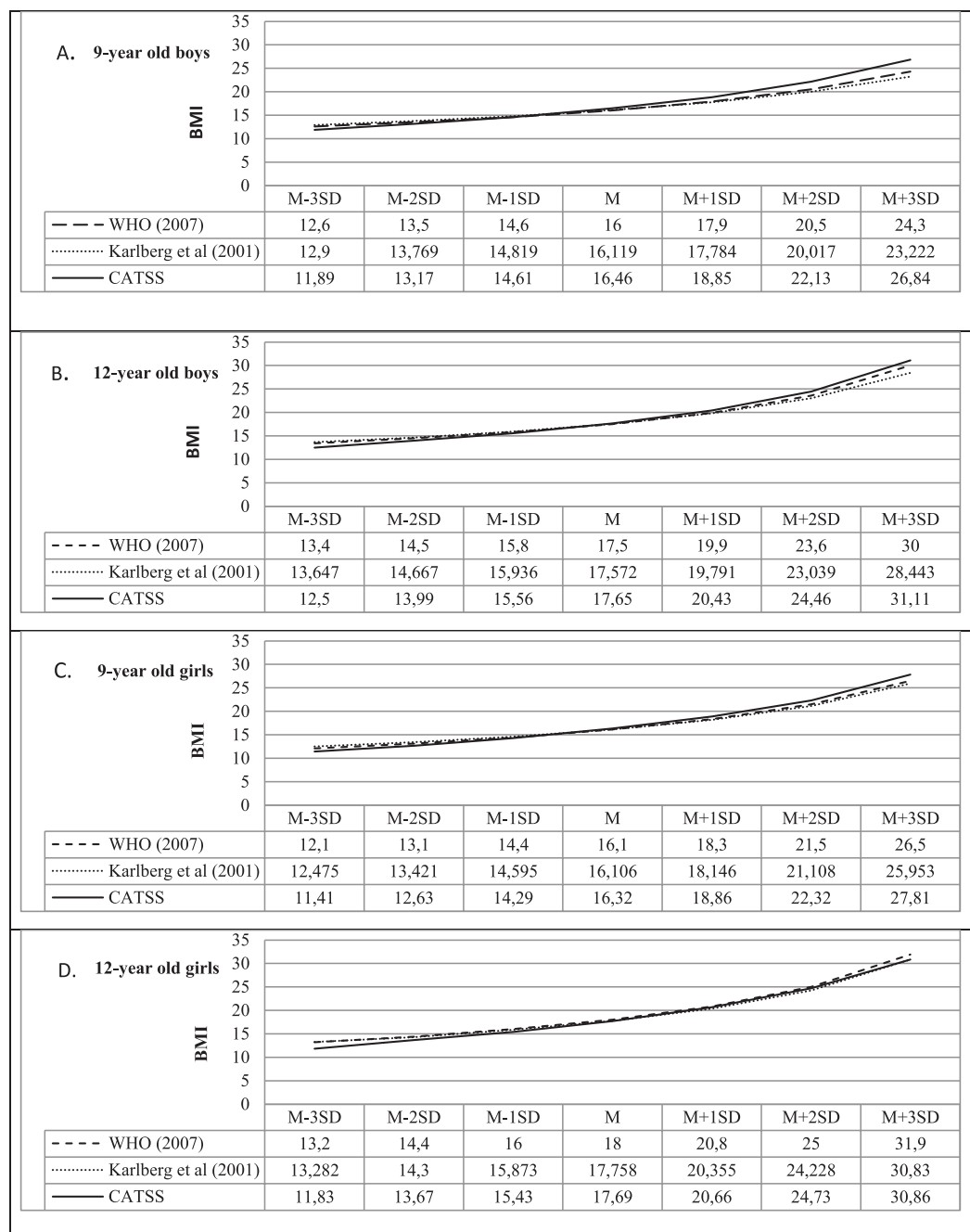

M = mean; SD = Standard deviation
BMI = Body mass index; CATSS = Child and Adolescent Twin Study in Sweden
WHO = World Health Organization

**Figure 3** Comparison between BMI references.

## Prevalence of extreme BMIs in children with ADHD and/or ASD

In the present study, children who scored above the diagnostic cutoff for ASD were overrepresented on both extreme ends of BMI. Three times as many children with ASD as children with no ASD or ADHD had high extreme BMIs, and twice as many had low

extreme BMIs. BMI in the group of children with ADHD followed a close to normal distribution, and the coexistence of ADHD with ASD seemed less often associated with extreme BMIs than ASD on its own. Extremes in BMI have previously been observed in children with ASD identified as overweight (*Chen et al., 2010*; *Rastam, 2008*) and underweight (*Hebebrand et al., 1997*; *Mouridsen, Rich & Isager, 2002*). Selective eating has been shown to be more prevalent among children with ASD, but no definite association has been found between selective eating and alterations in BMI among children with ASD (*Evans et al., 2012*), although restrictive eating in Canadian children 5 to 12 years old was associated with low BMI in one study (*Pinhas et al., 2011*).

## ADHD and ASD domains associated with extremes in BMI

The results of the present study show that the presence of ADHD is associated with a high extreme BMI, while ASD is associated with both the high and the low extremes of BMI.

While there are findings showing that impulsiveness in healthy adult females and males is associated with overeating (*Davis et al., 2006*; *Strimas et al., 2008*), an earlier study suggested that inattention is a reason for both low and high extreme weights (*Cortese & Vincenzi, 2012*; *Davis et al., 2006*). Our results may support the earlier study, that is, that the ADHD subdomain of inattention rather than the subdomain of impulsiveness/hyperactivity is associated with an increase in BMI. While ADHD scores were associated with a high extreme BMI, ASD scores were associated with both low and high extreme BMIs. Even in the presence of ADHD, the effect of ASD remained a significant risk factor for extremes of BMI in both directions. According to earlier studies, overweight and obesity are common in ASD, though probably not in children before the age of 7 (*Emond et al., 2010*), but increasingly so from school-age on (*Chen et al., 2010*). Underweight has also been reported in Danish and German adolescent boys with ASD (*Hebebrand et al., 1997*; *Mouridsen, Rich & Isager, 2008*; *Sobanski et al., 1999*).

## Etiology of BMI

As previously shown, the variance in children's BMI is largely influenced by genetic factors (*Dubois et al., 2012*). However, in younger children there is also a significant effect of shared environment (most easily translated as the family influence on eating behaviors, reviewed by *Birch & Fisher* in *1998*). The present study and a recent review on the heritability of BMI (*Elks et al., 2012*) suggest that the shared environmental effects during childhood decrease with age, while genetic predispositions have a larger effect later in life.

Because ASD was overrepresented on both ends of the BMI categories, it was not possible to conduct bivariate analyses of NPDs and BMI. The interpretation of such a model would be difficult and potentially misleading since ASD in the high end of BMI might be a different construct than ASD in the low end. Even with the relatively large number of subjects included in this study, tetrachoric correlations between ASD/ADHD and high and low BMIs could not be performed because too few individuals fulfilled each category to allow any conclusion to be drawn. Therefore, the present study did not analyze the importance of common genetic and environmental factors in the expression of NDPs and extreme BMI.

## Twin BMI standard

Because twins are generally smaller than singletons at birth, special twin BMI references are recommended for following them during their first years of life (*Estourgie-van Burk et al., 2010*; *Van Dommelen et al., 2008*). The information on BMI in twins older than 3 years old compared with singletons is contradictory. While some studies show that twins' BMI may still differ from singletons' at the age of 17 (*Pietilainen et al., 1999*), others have found no discrepancies in BMI between twins and singletons (*Wilson, 1979*; *Estourgie-van Burk et al., 2010*).

Our study confirmed that in pre-pubertal children, the BMI of twin samples could be considered to be representative of the general population of children. However, we found discrepancies between the twins' and national and international BMI standards for extremes in under- and overweight. This could be explained by the large sample size, which could increase the visibility of those more rare extremes. Another explanation could be that this is an expression of time trends in the BMI development of children. A third explanation would be that although mean BMIs in twins seem to be similar to those of singletons, twins may be overrepresented at the extreme ends of the BMI curve. These findings emphasize the need for a more updated and relevant BMI reference when assessing BMI deviations among twins in research and clinical practice.

### Summary

Children with ASD, with or without coexisting ADHD, are more prone to have low or high extreme BMIs than children without ADHD or ASD. The present study suggests that the inflexibility of children with ASD might affect their eating habits and contribute to high or low extremes in BMI, while the inattention problems of children with ADHD are associated solely with high extreme BMI.

### Strengths and limitations

To our knowledge, we are among the first to analyze the distribution of ADHD and ASD in children at the extremes of BMI in a nationwide study population. We have seen that most of the children with ADHD or ASD have normal BMIs, but that the prevalence of children at the extreme ends of the BMI curves was significantly higher among those with NDPs than in children without NDPs; that is, NDPs seem to carry increased risks for extreme BMIs. One obvious limitation is that the children were not clinically assessed and all data pertaining to NDPs and BMI were parent-reported. The accuracy of parentally reported weights and heights was assessed through comparisons with national and international standards of BMI (Result 1 and Appendix S1), and the parent interview used in our study to assess NDPs has shown excellent psychometric validity previously (*Hansson et al., 2005*; *Larson et al., 2010*). Another important limitation of our study that should be highlighted is the fact that the data collected concerned twins; although the literature shows conflicting results (*Estourgie-van Burk et al., 2010*; *Pietilainen et al., 1999*; *Van Dommelen et al., 2008*; *Wilson, 1979*), we suggest that twins at ages 9 and 12 do not differ as much as expected in their BMI from the general population.

*Clinical implications*

Based on our results we suggest that BMI references constructed for singletons may also be used in the clinic to recognize extremes of BMI in pre-pubertal twins.

Severe overweight and severe underweight should always be recognized in the assessment of children because of its impact on current and future health. Similarly, in children with ADHD, and even more so in those with ASD, the increased risk for both overweight and underweight should be considered.

### Funding

The CATSS-9/12-study is supported by the Swedish Council for Working Life and Social Research and the Swedish Research Council (Medicine). This publication was supported by Wilhelm och Martina Lundgren's Scientific grant Drn:81/2014. The funders had no role in study design, data collection and analysis, decision to publish, or preparation of the manuscript.

### Grant Disclosures

The following grant information was disclosed by the authors:
Swedish Council for Working Life and Social Research and the Swedish Research Council (Medicine).
Wilhelm och Martina Lundgren's Scientific: Drn:81/2014.

### Competing Interests

The authors have no conflicts of interest including financial interests and relationships and affiliations relevant to the subject of this manuscript. Nóra Kerekes and Sebastian Lundström are employed of the Swedish Prison and Probation Service.

### Author Contributions

- Nóra Kerekes and Armin Tajnia analyzed the data, wrote the paper, prepared figures and/or tables, reviewed drafts of the paper.
- Paul Lichtenstein and Henrik Anckarsäter conceived and designed the experiments, performed the experiments, reviewed drafts of the paper.
- Sebastian Lundström analyzed the data, prepared figures and/or tables, reviewed drafts of the paper.
- Thomas Nilsson wrote the paper, reviewed drafts of the paper.
- Maria Råstam conceived and designed the experiments, performed the experiments, wrote the paper, reviewed drafts of the paper.

### Human Ethics

The following information was supplied relating to ethical approvals (i.e., approving body and any reference numbers):

The CATSS study is in agreement with the Helsinki declaration and has been ethically approved by the Karolinska Institutet Ethical Review Board (Dnr 03-672 and 2010/507-31/1). All subjects have signed an informed consent that could be freely withdrawn at any time or at any phase of the study. All data analyses were performed on anonymized data.

## Data Deposition

The following information was supplied regarding the deposition of related data:

Raw data can be made available upon request to professor Paul Lichtenstein at the Swedish Twin Registry: http://ki.se/en/research/contact-the-swedish-twin-registry.

## Supplemental Information

Supplemental information for this article can be found online at http://dx.doi.org/10.7717/peerj.1024#supplemental-information.

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
