# Peer review of "Neurodevelopmental problems and extremes in BMI"

_PeerJ, doi:10.7717/peerj.1024_

## Round 0.1 · original submission · Major Revisions

Dear authors,

Please make sure to address the inspired comments of the two reviewers.

Reviewer 1 ·

Basic reporting

Results are properly reported.

Experimental design

The experimental design is sound

Validity of the findings

Findings are valid. The assessments used in the study measure the clinical manifestation of interest.

Additional comments

Authors study the association between ASD, ADHD and BMI in a large twin cohort.
They confirm results on the well known co-morbidity BMI and neurodevelopmental disorders.
The major comment pertains to the analysis of the twin data that they analyze as a regular cohort. How do authors account for the inter-correlation between ASD and BMI measures between twins ? This is not discussed in the methods. They don’t use the power of this large twin cohort. For example, they don’t use concordance rate of any of their measures in the analyses. This may help understand the directionality between these symptoms.


In the Introduction:

Authors state that “…Given the nature of the present study, we used only data
representing the ages 9 (n=828 boys & n=790 girls) and 12-year-old (n=640 boys & 676 girls)
children from Karlberg BMI reference (Karlberg et al. 2001)…”
This needs to be more specific. : Given the geographic origin and the age range of the cohort …

Authors state that “…The present information on BMI in twins (older than 3 years old)
compared to singletons is contradictory (Estourgie-van Burk et al. 2010; Pietilainen et al.
1999; Wilson 197)…”
This sentence needs clarification and brief introduction.

Authors state that “…Over the last few decades, the prevalence of ADHD seem to have
increased from 2%–5% (Kadesjo & Gillberg 1998) to 7%–9% (Scahill & Schwab-Stone
2000) in school-aged children, while the prevalence of autism increased from 0.04% (Lotter
1966) to 1.0% (Baird et al. 2006) for the broader spectrum of ASDs…”

For ASD, there is no evidence that the prevalence increased. Authors should handle this concept with caution. This should be replaced by “…diagnoses of ASD have increased…”

Authors state that “…An association between the two trends, namely the increase in BMI and the increased frequency of NDPs, has previously been suggested and investigated in a few studies, mostly in adults. Their conclusions are contradictory…”
Authors need to be very clear on the literature data. There are many studies on obesity and neuropsychiatric disorders occurring as co-morbidities, but there is no convincing data on the association between the increase of BMI and the increase of neuropsychiatric disorders. The papers that authors cite study comorbidity. This should be modified.

Methods:
What is the proportion of homozygous versus fraternal twins ?

In the Aims :
There is only lines on aim one in the introduction. There is a whole page related to aim 3 in the introduction. Aims and introduction should be prioritized accordingly.

Figure1 Is important but is not the main finding. Should be moved to supplemental data. Or a much more compact version should be designed for the main text.

Results:

The major comment pertains to the analysis of the twin data. How do authors account for the inter-correlation between ASD and BMI measures between twins ? This is not discussed in the methods.
They have access to a very large twin cohort but do not use concordance rate of any of their measures in the analyses. This may help understand the directionality between these symptoms.

Authors should perform analyses on the BMI acceleration between 9 and 12 in the different groups.

Authors should investigate the effect of the “gender x ASD measure” interaction term on BMI in table 2.

Given that this is a twin cohort, it is puzzling that the authors did not study the twin concordance rate of BMI, ASD and BMI-ASD as a comorbidity. It would add a significant element to the study.


Figure 2:
The n should be given for the different BMI categories on the x axis. In particular, extreme low BMI with ADHD or ASD amount to small numbers.

Discussion :

Authors should be cautious and only use the term “association”. The directionality of these associations is unknown. In particular, authors state that “…inattention is a reason for both low and high extreme weights…”

Authors should discuss the age effect.

Reviewer 2 ·

Basic reporting

No comments

Experimental design

Please see "General comments for the author" section

Validity of the findings

Please see "General comments for the author" section

Additional comments

In their manuscript entitled "Neurodevelopmental problems and extremes in BMI", Kerekes and colleagues investigate the relationship between BMI and ADHD and ASD in a dataset derived from parental telephone interviews of ~12,000 twins (ages 9 and 12). The study is important given that the question of the relationship between BMI and neurodevelopmental disorders is an interesting one, however I have a number of comments which I believe need to be addressed:

Major points

1. The authors do not justify their selection of BMI z-score cut-offs for low/high and extreme BMI. The choice of cut-offs should be justified, as this may influence the results of the study. How robust are the results to changes in the cut-offs used?
2. Given that the study is reliant on information supplied by the parents in telephone interviews, have the authors assessed whether accuracy of reporting of weight/height is uniform across the different categories (i.e. it may also be possible that height/weight measurements reported by parents of children with neurodevelopmental disorders may be more inaccurate)? If accuracy of reporting is not uniform across the categories this may influence the results of this study.
3. The authors do not specify whether children on medication which may influence weight were excluded from the study. This should be specified.
4. In the Results: (1) Twins BMI standard compared to singletons section, the authors state: "However, in our study generally extreme underweight and extreme overweight (with the exception of 12-year old girl's population) are more extreme." More details should be provided here.
5. In the Results section (2), P-values should be provided throughout this section.

Minor points

1. The introduction seems unnecessarily long, and should be kept more succinct.
2. There are a number of errors in grammar and vocabulary throughout the manuscript, and so it would benefit from being edited by a native English speaker.

---

## Round 0.2 · accepted · Accept

The reviewer's comments have been well taken care of and the manuscript is now suitable for publication.

With my best wishes